# Microbial Dynamics and In Vitro Degradation of Plant Secondary Metabolites in Hanwoo Steer Rumen Fluids

**DOI:** 10.3390/ani11082350

**Published:** 2021-08-09

**Authors:** Dahye Kim, Palaniselvam Kuppusamy, Jeong Sung Jung, Kyoung Hoon Kim, Ki Choon Choi

**Affiliations:** 1Faculty of Biotechnology, College of Applied Life Science, Jeju National University, Jeju 63243, Korea; pioioiq10@gmail.com (D.K.); jjs3873@rda.go.kr (J.S.J.); 2Grassland and Forage Division, National Institute of Animal Science, Rural Development Administration, Cheonan 31000, Korea; kpalaselvamsmailbox@rediffmail.com; 3Department of International Agricultural Technology, Graduate School of International Agricultural Technology, Seoul National University, Pyeongchang 25354, Korea; khhkim@snu.ac.kr; 4Department of Eco-Friendly Livestock Science, Institutes of Green Bio Science and Technology, Seoul National University, Pyeongchang 25354, Korea

**Keywords:** rumen, plant secondary metabolites, degradations, HPLC-DAD, microbial dynamics

## Abstract

**Simple Summary:**

In the current study, we investigated changes in the levels of plant secondary metabolites (PSMs) in a rumen buffer in vitro model and identified the microbial dynamics of the rumen sample. The data suggested that the different types of PSM including luteolin, ferulic acid, caffeic acid, coumaric acid, rutin, myricetin, vitexin, kaempferol, and quercetin were decreased after 12 h of incubation in the rumen fluid (*p* ≤ 0.05). In contrast, the propyl gallate concentration was not significantly changed even after 24 h of incubation in rumen fluid compared to other metabolites. Microbial dynamics study showed that the Firmicutes, Bacterodetes, Actinobacteria, and Syngergistetes were the dominant phyla found in rumen fluids. Overall, data suggested that most polyphenolic compounds may degrade or reform new complex structures in the rumen. Further in vivo studies are needed to determine the PSM conversion rate and the final products from native PSM when undergoing degradation or reforming new complexes, and the effect on the microbiome.

**Abstract:**

Plant secondary metabolite (PSM) degradations and feed breakdown into small particles may occur primarily in the rumen. It is possible to predict the rate and extent of feed disappearance in the rumen during incubation by different in vitro techniques, which differ based on the PSM structures, including phenolics, and flavonoids. However, PSM degradation and conversion efficiency in the rumen remains unclear. This study’s objective was to evaluate the in vitro degradation of a group of PSMs in the rumen fluid, collected from Hanwoo steer samples. PSMs including rutin, vitexin, myricetin, p-coumaric acid, ferulic acid, caffeic acid, quercetin, luteolin, propyl gallate, and kaempferol were used in their pure forms at 1mg/250 mL in a rumen fluid buffer system. The mixture of selected PSMs and buffer was incubated at 39 °C for 12–72 h, and samples were collected every 12 h and analyzed by a high-performance liquid chromatography-diode array detector (HPLC-DAD) to determine the biotransformation of the polyphenolics. The results revealed that the luteolin, ferulic acid, caffeic acid, coumaric acid, rutin, myricetin, vitexin, kaempferol, and quercetin were decreased after 12 h of incubation in the rumen fluid (*p* ≤ 0.05) and were more than 70% decreased at 72 h. In contrast, the propyl gallate concentrations were not significantly changed after 24 h of incubation in rumen fluid compared to other metabolites. Finally, microbial dynamics study showed that the Firmicutes, Bacterodetes, Actinobacteria, and Syngergistetes were the dominant phyla found in rumen fluids. The data suggest that most polyphenolic compounds may degrade or reform new complex structures in the rumen.

## 1. Introduction

Polyphenols are an extensive class of secondary plant metabolites that exert beneficial effects on ruminant nutrition. In addition, these secondary metabolites can suppress harmful microbial activity and enhance valuable dietary protein degradation in the rumen through their potent antimicrobial and antioxidant properties [1]. Interest into research on the effects of polyphenols on animal nutrition has significantly increased. These compounds are ubiquitous in the plant kingdom, particularly in plants exploited as alternative feeding resources to cultivated crops and many agro-industrial by-products. Various polyphenols are found among secondary plant metabolites, ranging from simple phenolic molecules to highly polymerized compounds. This modulates ruminal metabolism [2], growth performance, health status [3], and product quality [4].

Soy (*Glycine max*), Italian ryegrass (*Lolium multiflorum* L.), alfalfa (*Medicago sativa* L.), and red clover (*Trifolium pratense* L.) are major sources of phenolics and isoflavones that are commonly used in the feed of dairy cattle. These forages are a rich source of isoflavones and other metabolites that can be further degraded by microorganisms in the rumen to become more potent energy sources [5]. Similarly, phenolic acid derivatives are potent bioactive metabolites due to their antimicrobial and anti-inflammatory properties, although their nutritional physiology has not yet been defined. However, when claiming biological activities of a secondary plant metabolite, it is essential to understand its digestion, biotransformation, absorption, and distribution in animals and humans [6]. Absorption, metabolism, deposition, and excretion are strongly dependent on the physicochemical characteristics of the compounds involved and their susceptibility to transformation. In addition, when ingested, relevant plant secondary metabolites (PSMs) may pass through the animal as non-metabolized substrates or combine with bile salts and become discharged in the feces [7]. Sinz et al. reported flavonoids such as tannic acid, epicatechin, quercetin, isoquercetin, and luteolin-7-glucoside reduced the amount of methane produced in relation to total gas produced. The flavonoids tested, especially luteolin-7-glucoside seemed to have a similar potential as tannic acid to mitigate methane and ammonia formation during ruminal fermentation in vitro, both favorable with respect to the environment [8]. In addition, probiotics or the combination of yeast and phytochemicals, mainly flavonoids, could be compatible with each other and improve rumen fermentation, but supplementation with *C. tropicalis* had limited effects on increasing growth performance and decreasing fecal scores compared with flavonoid supplementation. Therefore, the flavonoids have an effect on health or rumen fermentation in pre- and post-weaning calves [9].

The rumen is the largest compartment of the ruminant digestive tract. It contains complex microbial ecosystems, including bacteria, protozoa, and fungi. These microbial communities play an important role in the digestion, absorption, and fermentation of ingested feedstuffs in the rumen [10]. Rumen microorganisms can convert complex secondary plant metabolites into fermented end products, such as simple sugars and volatile fatty acids, which the ruminant host can utilize for their energy requirements. Plant metabolites also have direct effects on ruminant health and nutrition, as well as methane gas production [11]. The degradation and metabolism of carbohydrates and proteins can be accomplished by microbial enzymes present in the rumen. However, the structural complexity, insolubility, and initial inaccessibility of the cell wall components in rumen often confine the extent to which they are fermented in the rumen [12]. The rumen is a typical host–microbe symbiotic system that possesses prominent cellulolytic bacterial populations, major fibrinolytic anaerobic fungi, and protozoa. These diverse microbial communities are involved in the digestion of plant-based feedstuffs and their subsequent conversion for energy requirements of the host ruminant. Therefore, rumen symbiotic microbes play an important role in the digestion and/or degradation and absorption of nutrients [13]. The current study evaluated whether polyphenolic compounds undergo degradation/changes in rumen fluids of Korean native cattle (Hanwoo steers) in vitro. Furthermore, we assessed microbial dynamics in rumen fluids by a pyrosequencing tool.

## 2. Materials and Methods

### 2.1. Experimental Design, Animals, and Diet

All experimental procedures and animal care protocols used in this study were approved by the Animal Care Committee of the National Institute of Animal Science, Korea (Code Number: NIAS 2019-357). Briefly, this study was carried out using three ruminal cannulated Hanwoo steers with a live weight of around 580 ± 5 kg (30 ± 2.5 months old). During the experimental period, each Hanwoo steer received a diet of forage with concentrate at 12 kg per day with free access to water. The concentrate and forages were present at a ratio of 2.5:7.5, respectively. The experimental animal diet supplementation is shown in Table 1.

### 2.2. Fermentation Media Preparation

Ruminal fluid was collected from fistulated Hanwoo steers (three Hanwoo steers and collected separately) four hours after feeding and placed into a sterilized thermos bottle for transport to the laboratory. The ruminal content was mixed well with CO_2_ gas for 1 min to liberate any additional fiber attached to micro-organisms. The mixed rumen fluid was strained through four layers of cotton cloth to collect the inoculum for the in vitro fermentation study. The buffer solution was made according to Holden, (1999) and consisted of solutions A and B. Buffer A solution contained KH_2_PO_4_ (10.0 g), MgSO_4_·7H_2_O (0.5 g), NaCl (0.5 g), CaCl_2_·2H_2_O (0.1 g), and urea (0.5 g) to 1000 mL of distilled water and buffer B solution contained Na_2_CO_3_ (15 g) and Na_2_S·9H_2_O (1 g) to 1000 mL of distilled water [14]. Solution A was mixed with solution B at a ratio of 5:1 to make 1600 mL of buffer solution and was mixed well with 400 mL of rumen fluid. Then, it was combined well with CO_2_ gas for 15 min before dispersing. The buffer solution provided an obligatory anaerobic environment during the fermentation process. For the secondary metabolite degradation study, 3500 mL glass bottles (Ankom Technology, Macedon, NY, USA) were used for the fermentation procedure.

### 2.3. In Vitro Degradation of Polyphenolics Using Daisy Incubator

The degradability of phenolics acids and flavonoids in the rumen fluid was determined using an in vitro digestibility technique with slight modifications [14]. In vitro digestibility was analyzed with a little modification and a methodological change proposed by Ankom Technology Corporation for its DaisyII^®^ incubator (Model D220, Macedon, NY, USA). Each 1 mg of phenolic and flavonoid sample was placed into a 500 mL Erlenmeyer flask containing 250 mL of rumen fluid buffer system and incubated at 39 °C with shaking (30 oscillations/min). The Ankom Technology DaisyII^®^ incubator allows for the simultaneous incubation of up to 10 different phenolic and flavonoid samples and buffer placed in glass containers at a uniform temperature and under constant agitation during the incubation process [15]. Each pure phenolic and flavonoid (1 mg) sample was prepared and added together in the fermentation vessel containing buffered inoculums. The addition of these metabolites to this in vitro model was used to determine the hydrolysis of the selected phenolics and flavonoids at different time intervals compared to 0 h using HPLC-DAD (Agilent Technologies, model-1100, Santa Clara, CA, USA). These samples were incubated at 39 °C for 12–72 h to quantify of the amount of degradation of pure phenolics and flavonoids during the rumen fermentation period. At each time, the fermented rumen samples were carefully collected using a 5 mL sterile syringe to evaluate the hydrolysis of the polyphenolic compounds (number of samples, three; technical replicates, each, three; and number of HPLC runs, two)

### 2.4. Detection of Phenolic and Flavonoid Degradation by Rumen Fluid by HPLC-DAD Analysis

HPLC-DAD analyses were carried out according to a method described previously [16]. After incubation in rumen fluid, samples of polyphenolic compounds were collected at 12 h intervals, centrifuged at 5000 rpm for 10 min at 4 °C, and filtered through a 0.45 µm GHP membrane filters (Life Science PALL, Washington, NY, USA). HPLC analyses were performed using an 1100 Agilent Series system consisting of a binary pump, an online degasser, and an auto-sampler. A YMC-triart C18 (150 × 4.6 mm, 12 nm) column was used to separate all substances and maintained at ambient temperature. Sample peaks (including standard phenolics and flavonoids) were identified by comparison to known standards. The HPLC mobile phase, a binary gradient system consisting of (A) 0.1% formic acid in water and (B) acetonitrile, was used at a flow rate of 0.8 mL/min with the following schedule: 2% B at 0–1.5 min, 50% B at 15 min, 90% B at 15.1–20 min, and 2% B at 20.1 –25 min. The injection volume was 10μL. Metabolites were detected at DAD and UV wavelengths of 240 and 360 nm, respectively.

### 2.5. Microbiome Study by Pyrosequencing Analysis

The rumen microbiota compositions were analyzed by quantitative PCR (qPCR) and pyrosequencing techniques. These molecular methods could estimate the numbers of microbiota in the rumen fluids. According to the manufacturers’ protocol, total genomic DNA was extracted from the pellets derived from 1.5 mL of rumen filtrate using a PowerSoil^®^ DNA Isolation Kit (MO BIO, Seoul, Korea). The DNA quality and quantity were measured by PicoGreen and a Nanodrop machine, respectively. The 16S rRNA genes were amplified using 16S V3-V4 primers. The primer sequences were as follows: Forward Primer-5’TCGTCGGCAGCGTCAGATGTGTATAAGAGACAGCCTACGGGNGGCWGCAG. Reverse Primer-5’GTCTCGTGGGCTCGGAGATGTGTATAAGAGACAGGACTACHVGGGTATCTAATCC. The gDNA was amplified with 16S V3-V4 primers, and a subsequent limited-cycle amplification step was performed to add multiplexing indices and Illumina sequencing adapters. The final products were normalized and pooled using PicoGreen, and the size of libraries was verified using a Tape Station DNA Screen Tape D1000 (Agilent). The products were sequenced using a MiSeq™ platform (Illumina, San Diego, CA, USA) by Macrogen, Seoul, Korea. [17].

### 2.6. Statistical Analysis

All numerical data were obtained from three replicative experiments. The data are presented as mean ± STD. The significant differences were analyzed by SPSS16 software (SPSS Inc., Chicago, IL, USA) using analysis of variance (one-way ANOVA, included post hoc, Duncan, and descriptive analysis parameters) followed by the least significant difference test. *p* values of less than 0.05 were considered statistically significant.

## 3. Results

### 3.1. Hydroxycinnamic Acid Status in Rumen Fluids after Various Incubation Periods

The concentration–time curves of hydroxycinnamic acids such as coumaric acid, ferulic acid, and caffeic acid in Hanwoo steer in vitro rumen fluid during incubation periods are presented in Table 2. During the first 12 h of incubation, coumaric acid and ferulic acid concentrations decreased significantly (*p* ≤ 0.05). However, the caffeic acid concentration was not altered at 12 h incubation. All the cinnamic acid levels were reduced after 12 h incubation; in particular, at 72 h, the range of cinnamic acids was rapidly reduced.

### 3.2. Flavonol and Flavone Changes in Rumen Fluids at Different Incubation Periods

The concentrations and time curves of flavonols such as myricetin, quercetin, and kaempferol and the flavone luteolin in rumen fluids at different incubation periods are presented in Table 3. The estimated concentration of flavonols and flavone in rumen fluids remained unchanged during the first 12 h of incubation. In contrast, during incubation periods from 12 h to 72 h, the concentrations of myricetin, quercetin, and kaempferol rapidly decreased. However, compared to other flavonols, luteolin had virtually disappeared from the fluid after 12 h incubation.

### 3.3. Flavonol Glycosides and Ester Derivative Changes in Rumen Fluids at Different Incubation Periods

The concentrations of flavonol glycosides such as rutin, vitexin, and the ester derivative propyl gallate in rumen fluids at different time intervals are presented in Table 4. The concentration of rutin and vitexin remained unchanged in rumen fluid at 12 h incubation, but after 12 h of incubation, the rutin and vitexin were rapidly reduced in rumen fluid. Propyl gallate showed no significant changes at 12 and 24 h of incubation in Hanwoo steer in vitro rumen fluid.

### 3.4. Microbial Community Dynamics of Rumen Fluid

Animal nutritionists have been focused on the rumen’s physiology and its microorganisms and their role in digestion and utilization. It is widely accepted that feed utilization in the rumen is based on the rumen’s whole microbiota. Therefore, we planned to investigate the rumen’s microbial dynamics using a pyrosequencing tool. The results revealed that the mean reads were 40,249 ± 2631.9 and similar OUT (578.16 ± 26.8), Cho1 (614.7 ± 24.2), and Shannon (6.59 ± 0.10) was noted in all three rumen fluids. This suggests that the microbial diversity and richness of the samples were similar. Firmicutes (66.35%) was dominant phylum in the ruminant fluids followed by Bateroidetes (12.38%), Actinobacteria (1.13%), Synergistetes (1.12%), Euryarchaeota (0.41%), and Proteobacteria (0.32%) (Figure 1A). Genus level analysis revealed that Intestinimonas, Ruminococcus, Christensenella and Prevotella were the dominant genera found in the rumen (Figure 1B) fluid. At species level, *I. butyriciproducens* (14.9%), *L*. *thermophila* (3.66%), *F. butyricus* (3.06%), *C. massilliensis* (2.91%), and *P. ruminicola* (2.21%) were the dominant species found in the rumen fluids (Figure 1A–C).

## 4. Discussion

Rumen fluid has been used as an incubation inoculum with a fermentation medium to mimic in vivo digestibility [18]. PSMs can serve as an alternative for improving animal performance without compromising safety aspects. Most PSMs can be served as natural resources for animal production. In recent years many researchers emphasized that a PSM group can influence rumen fermentation favorably, so it can be used as an alternative for improving ruminant productions. However, the role and degradation of PSMs in rumen fluid remains unclear. In this study, we aimed to determine the PSM stability/degradability in rumen fluids after certain incubation periods. Generally, isoflavones and natural compounds were metabolized and their concentrations were decreased during in vitro incubation. The researcher stated that equol production began after 3h incubation and reached a maximum level after approximately 12h. The equol production was continued even after 24h incubation, unless the daidzein concentration decreased [5]. A wide range of bacterial populations present in the rumen responsible for the esterolysis of conjugated hydroxycinnamic acids have been identified. In addition, results have confirmed that selected strains of *Escherichia coli*, *Bifidobacterium lactis*, and *Lactobacillus gasseri* have esterase activity with the ability to cleave chlorogenic acid (C-QA) and ethyl ferulate [6]. Previous reports have claimed that phenolic acids, such as cinnamic acid, undergo degradation when they are incubated with rat cecal medium [19]. The hydrolysis of flavonoid monomers by mixed microbes from adapted sheep, non-adapted sheep, and goats has been observed by HPLC analysis. Flavonoid peaks do not only indicate the complete degradation of tannins in the feedstuff as many unknown peaks can appear after incubation with the feedstuff. These peaks might also come from the hydrolysis of polymeric phenolic compounds and their monomers. In addition, rumen fluid contains various mixed cultures that could degrade pure phenolic compounds into monomers such as tannic acid or gallic acid, and these monomers can be observed in HPLC chromatograms [20]. Further studies are needed to confirm the hypothesis that mixed rumen microbiota from those animals that have adapted to tannin-rich feed could convert tannic acid to simple monomers.

Secondary plant metabolites, such as phenolics, flavonoids, saponins, tannins, essential oils, and organosulphur-containing compounds, are involved in the productivity and health of livestock animals. These metabolites can also control nutritional stress, such as acidosis and methanogenesis formation, in the rumen. The rumen pH values were slightly increased in flavonoid-rich supplemented diet forage crops compared to those of the control group. This might be due to the beneficial effect of flavonoids in enhancing the lactate-consuming microorganisms of the rumen. Three forage crops metabolites (*Acacia nilotica, Ziziphus nummularia,* and *Vigna sinensis*) were degraded differently in the in vitro ruminal fermentation system. The substrates from *A. nilotica* feedstuff were hydrolyzed at a higher rate than other plant extracts [21]. Authentic cultures of rumen microbiota were different in their tolerance of phenolic acids [22]. Hemicellulolytic populations are known to be more tolerant of phenolics than cellulolytic bacteria because phenolic acids can bind to hemicellulose materials of the forage cell wall. This suggests that bacterial species that are exposed to high concentrations of phenolic compounds under natural conditions might have evolved higher tolerance to these compounds [23].

Plant polyphenols have an effect on the rumen microbiota responsible for unsaturated fatty acid biohydrogenation, fiber digestion, and methane production [24]. High amounts of tannins (both condensed tannins (CT) and hydrolyzable tannins (HT)) significantly reduce the voluntary feed intake and animal performance, compared to low or medium doses. This depends on the heterogeneity of the dietary polyphenolic composition. It is possible to state that low to moderate amounts in the diet of dairy ruminants had detrimental effects on animal performance. Previous studies reported the differential effects of phenolics on the components of the microbial population, including particular bacterial species or taxonomic groups of microorganisms, such as the fungi and protozoa, indicating that they may not be equally susceptible to the effects of specific phenolics. This factor may have led to the differences in the observed concentrations and proportions of volatile fatty acids (VFAs) produced in the presence of the phenolics. In contrast, the fermentation of arbutin produced contrasting patterns of VFAs compared to orcinol and quinol, which decreased acetate and propionate concentrations relative to the controls but stimulated butyrate production [25].

The common flavonoid ring system (e.g., quercetin and kaempferol) and phenolic glycosides, such as rutin, are primarily degraded in the rumen and intestinal microbiota by the cleavage of the hydroxycylic ring and the hydrolysis of acetate, butyrate, monohydroxy phenolics, and phloroglucinol [26]. In addition, the hydrolyzable tannins are readily degraded by tannase enzymes of the rumen bacteria and fungi that degrade the galloyl residues of galloyl esters, hexahydroxy diphenoyl groups, and other ellagitannin derivatives. Tannase-positive microorganisms have been reported in cattle, sheep, goats, deer, koala, and humans consuming tannin-containing diets. In addition, phloroglucinol is a synthetic phenolic compound formed through the degradation of complex plant molecules, such as flavonoids, which are commonly present in the diet of grazing ruminants [27]. Phenolic compounds did not affect the total digestibility of roughage or dry matter, nutrient content, or microbial protein synthesis efficiency. Phenolic compounds reduced the *Entodinium* protozoan population in water buffalo and changed the ruminal fermentation pattern, eventually favoring acetate fermentation [28]. The *Coprococcus* genus was mainly involved in key metabolic pathways in the rumen, and might improve the rumen efficiency and degrade plant toxins in ruminants [29]. The ruminal degradation of quercetin with its influence on ruminal gas production using concentrate, grass hay, and straw as feed substrates studied in vitro [30]. Quercetin was rapidly degraded and the disappearance of quercetin was accompanied by the simultaneous appearance of two monomers—3, 4-dihydroxy-phenylacetic acid and 4-methyl catechol. Reduction in propyl gallate concentration was noted after 24 h incubation. We hypothesized that the rate of decrease in individual PSMs appeared to be related to the groups of phenolic acid and flavonoids digested in the plant-based feeds. The estimated mean initial concentration of the phenolics and flavonoids was lower than the final concentration of each phenolic and flavonoid compound. However, a statistically significant difference was only observed for luteolin, ferulic acid, coumaric acid, quercetin, and myricetin. An in vitro method has different physicochemical and biological factors compared to an in vivo environment that influences the degradation or conversion of phenolic and flavonoid compounds in the rumen. The in vitro degradation of the substrate may be affected by several factors, such as the stirring speed and solid dilution rate, which can affect the metabolites’ degradability based on HPLC-DAD retention time. Plant derivatives, such as phenolics and flavonoid compounds, could be partially hydrolyzed by the mixed culture from the rumen microbial biomass. Some PSM might remain in the in vitro rumen fluid in the form of monomers without degradation.

In contrast, the flavonoids, such as myricetin, catechin, rutin, flavone, and kaempferol, at concentrations of 4.5% in a dry matter diet reduced the population of rumen microbes, while naringin and quercetin decreased protozoa and methanogen populations in vitro [18]. Previously, several reports have illustrated that flavonoids may regulate ruminal microbiota and that different types of flavonoids from various sources have distinct effects. Similarly, the ruminal microbial biomass can be influenced by in vitro incubation with different plant-derived substrates [31]. The proportion and composition of microbial flora are also influenced by feed types, diet, feeding environment, age, climate, and so on. Furthermore, the ecological community of the microbiota in the rumen can directly affect the fermentation pattern of the feed substrates. Hence, the digestion and absorption of feedstuff highly depend on bacterial flora changes in the rumen. In addition, metabolites can undergo several biotransformations until they are excreted through the urine or the bile. Finally, non-degraded complex secondary plant metabolites can be re-absorbed during intestinal digestion and excreted through the feces.

## 5. Conclusions

In conclusion, the present study examined the dietary phenolic and flavonoid degradation response in rumen fluid. Different groups of phenolics and flavonoids are degraded (12–72 h) by ruminal microbiota. The bioavailability of these metabolites may allow their transfer to other digestive compartments, such as duodenal and intestinal digestion. Rutin, coumaric acid, kaempferol, vitexin, quercetin, ferulic acid, caffeic acid, luteolin, and myricetin showed significant changes in ruminal fermentation processes. Thus, they can be considered harmless to the microbial community in the rumen. Further in vivo studies are needed to determine the feed conversion rate and the effect of the microbiome on the nutrition degradation status.

## Figures and Tables

**Figure 1 animals-11-02350-f001:**
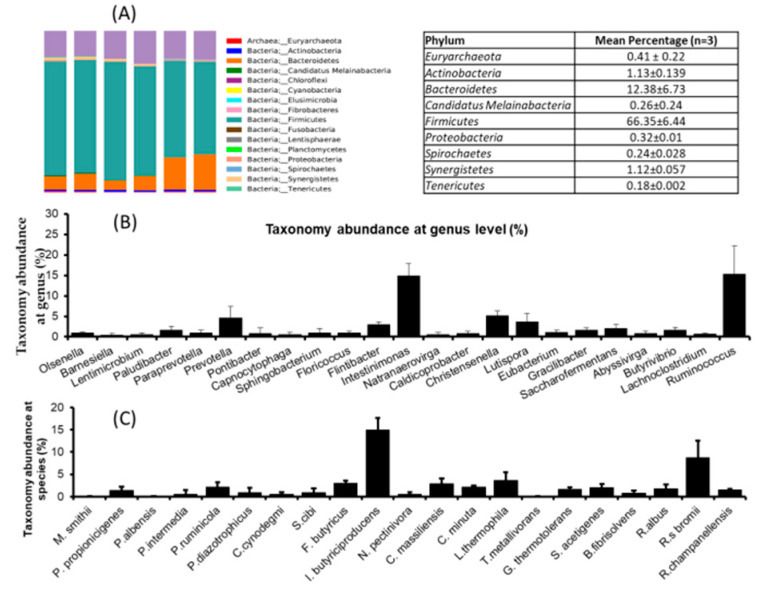
Microbial dynamics study of rumen fluids by pyrosequencing method. (**A**) Taxonomic abundances in rumen fluids at phylum level, (**B**) taxonomic abundances in rumen fluids at genus level, and (**C**) taxonomic abundances in rumen fluids at species level. The data are represented as mean ± STD, *n* = 3.

**Table 1 animals-11-02350-t001:** Diet supplementation used in this experiment.

Ingredients	% of DM
Rice straw	12.5
Timothy	12.5
concentrate	
Maize	30.46
Corn gluten feed	15.56
Wheat gluten	10.01
Soybean meal	7.37
Rapeseed meal	3.12
Coconut kernel meal	3.11
Palm kernel meal	3.11
Limestone	1.33
Salt	0.36
Sodium hydrogen carbonate	0.36
Vitamins and minerals *	0.21

* Grobig-DC provides the following nutrients per kg of diet: vitamin A, 2,650,000 IU; vitamin D3, 530,000IU; vitamin E, 1050IU; nicotinic acid, 10,000 mg; Fe, 13,200 mg; Mn, 4400 mg; Zn, 4400 mg; Cu, 2200 mg; iodine, 440 mg; cobalt, 440 mg. (Elanco Animal Health, Korea).

**Table 2 animals-11-02350-t002:** Quantification of hydroxycinnamic acids in rumen fluids after different incubation periods by HPLC-DAD method.

Incubation Time (h)	Coumaric Acid (µg)	Ferulic Acid (µg)	Caffeic Acid (µg)
0	178 ± 0.15 ^a^	164 ± 0.11 ^a^	167 ± 0.13 ^a^
12	142 ± 0.21 ^b^	142 ± 0.18 ^b^	172 ± 0.16 ^a^
24	112 ± 0.13 ^c^	132 ± 0.16 ^b^	143 ± 0.17 ^b^
48	90.0± 0.14 ^d^	110 ± 0.22 ^c^	109 ± 0.09 ^c^
72	54.0 ± 0.10 ^e^	88.0 ± 0.17 ^d^	68.0± 0.10 ^d^

The mixture of selected secondary metabolites was incubated at 39 °C for 12–72 h, and sampled at 12 h intervals for determination of polyphenolic concentrations in rumen fluid. The data are represented as mean ± STD, *n* = 3. ^abcde^ letters in the column indicate significant differences between incubation periods (*p* ≤ 0.05).

**Table 3 animals-11-02350-t003:** Quantification of flavonols in rumen fluids after different incubation periods by HPLC-DAD method.

Incubation Time (h)	Myricetin (µg)	Luteolin (µg)	Quercetin (µg)	Kaemferol (µg)
0	174 ± 0.23 ^a^	140 ±0.14 ^a^	174 ± 0.12 ^a^	188 ± 0.08 ^a^
12	178 ± 0.13 ^a^	133 ± 0.18 ^a^	184 ± 0.13 ^b^	179 ± 0.16 ^b^
24	123 ± 0.11 ^b^	ND	105 ± 0.08 ^c^	140 ± 0.14 ^c^
48	90.0 ± 0.07 ^c^	ND	70.0 ± 0.14 ^d^	102 ± 0.11 ^d^
72	71.0 ± 0.15 ^d^	ND	45.0 ± 0.12 ^e^	72.0 ± 0.15 ^e^

The mixture of selected secondary metabolites was incubated at 39 °C for 12–72 h, and sampled at 12 h intervals for determination of flavonol and flavone concentrations in rumen fluid. The data are represented as mean ± STD, *n* = 3. ^abcde^ letters in the column indicate significant differences between incubation periods (*p* ≤ 0.05); ND, not detected.

**Table 4 animals-11-02350-t004:** Quantification of flavonol glycosides and ester derivatives in rumen fluids after different incubation periods by HPLC-DAD method.

Incubation Time (h)	Prophyl Gallate (µg)	Vitexin (µg)	Rutin (µg)
0	168 ± 0.11 ^a^	173 ± 0.06 ^a^	176 ± 0.15 ^a^
12	171 ± 0.12 ^a^	179 ± 0.09 ^a^	177 ± 0.17 ^a^
24	164 ± 0.37 ^b^	119 ± 0.11 ^b^	106 ± 0.21 ^b^
48	140 ± 0.09 ^c^	81.0 ± 0.09 ^c^	78.0 ± 0.06 ^c^
72	110 ± 0.12 ^d^	66.0 ± 0.12 ^d^	39.0 ± 0.17 ^d^

The mixture of selected secondary metabolites was incubated at 39 °C for 12–72 h, and sampled at 12 h intervals for determination of flavonol concentrations in rumen fluid. The data are represented as mean ± STD, *n* = 3. ^abcde^ Alphabets in the column indicate significant differences between incubation periods (*p* ≤ 0.05).

## Data Availability

Data are available on request from the corresponding author.

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
