# Peer review of "Microbial Dynamics and In Vitro Degradation of Plant Secondary Metabolites in Hanwoo Steer Rumen Fluids"

_animals, 2021, doi:10.3390/ani11082350_

Round 1

Reviewer 1 Report

L29; Firmicutes, Bacterodetes, Actinobacteria and Syngergistetes
L43; metabolism[2],  status[3],  quality[4] --- metabolism [2],  status [3],  quality [4]
L62; respect[8] --- respect [8]
L93; water (NIAS 2007). --- Is this reference ? Is citation rule as [?] ?
L106; Holden, (1999)  --- Is this reference ? Is citation rule as [?] ?
L107; MgSO4.7H2O --- MgSO4・7H2O
CaCl2.2H2O --- CaCl2・2H2O
Na2S.9H2O --- Na2S・9H2O
L128; HPLC–DAD --- (Model No. name, city, country of company) Agilent Technologies ?
L132; (Number of samples 3, Technical --- (number of samples 3, technical
L164; mean means ± STD --- OK?
SPSS16 --- (name, city, country of company)
L215; 40249± 2631.9  (578.16± 26.8)  (614.7±24.2)  (6.59 ±0.10) --- 40249 ± 2631.9  (578.16 ± 26.8)  (614.7 ± 24.2)  (6.59 ± 0.10)
L220; (Fig.2A). L221; (Figure 2B) --- Which is OK ?
L223; I.butyriciproducens (14.9%),L.thermophila --- I. butyriciproducens (14.9%), L. thermophila
L314; In- vitro --- In vitro

Author Response

We thank the reviewer for giving valuable comments on our research paper. We have carefully gone through the whole manuscript and modified it accordingly. We have provided a pointwise response to all statements given by the reviewer. 

  1. L29; Firmicutes, Bacterodetes, Actinobacteria and Syngergistetes L43; metabolism[2],  status[3],  quality[4] --- metabolism [2],  status [3],  quality [4] L62; respect[8] --- respect [8]

 We have provided a space between reference number and word in the manuscript.

Phylum names were capitalized (Finally, microbial dynamics study showed that the Firmicutes, Bacterodetes, Actinobacteria and Syngergistetes were the dominant phyla found in rumen fluids)

  1. L93; water (NIAS, 2007). --- Is this reference? Is citation rule as [?] ?

Thanks, it is typographical mistakes, now we have deleted this word  from the text

  1. L106; Holden, (1999)  --- Is this reference ? Is citation rule as [?] ?

The reference for above said citation has provided (reference Number 14)

  1. L107; MgSO4.7H2O --- MgSO4・7H2O CaCl2.2H2O --- CaCl2・2H2O Na2S.9H2O --- Na2S・9H2O

Thanks, we have modified the molecular formula according to the reviewer suggestion

  1. L128; HPLC–DAD --- (Model No. name, city, country of company) Agilent Technologies ?SPSS16 --- (name, city, country of company)

We have provided company details for HPLC and SPSS in the method section.

HPLC–DAD (Agilent 1100, CA, USA)

SPSS16 software (SPSS Inc, IL, USA)

  1. L132; (Number of samples 3, Technical --- (number of samples 3, technical L164; mean means ± STD --- OK? L215; 40249± 2631.9  (578.16± 26.8)  (614.7±24.2)  (6.59 ±0.10) --- 40249 ± 2631.9  (578.16 ± 26.8)  (614.7 ± 24.2)  (6.59 ± 0.10) L223; I.butyriciproducens (14.9%),L.thermophila --- I. butyriciproducens (14.9%), L. thermophile  L314; In- vitro --- In vitro

Thanks, we have provided space between words and removed repeated word means as per the reviewer suggestion.

  1. L220; (Fig.2A). L221; (Figure 2B) --- Which is OK ?

Thanks and Sorry, The mistake has modified   Line No: 222-228  It suggests that the microbial diversity and richness of the sample were similar. Firmicutes (66.35%) was dominant phylum in the ruminant fluids followed by Bateroidetes (12.38%), Actinobacteria (1.13%), Synergistetes (1.12%), Euryarchaeota (0.41%) and Proteobacteria (0.32%) (Figure 4a). Genus level analysis revealed that Intestinimonas, Ruminococcus,  Christensenella and Prevotella were the dominant genera found in the rumen (Figure 4b) fluid

Reviewer 2 Report

The manuscript has been improved both from a methodological point of view and in the discussion of the results.

Author Response

The manuscript has been improved both from a methodological point of view and in the discussion of the results.

We thank the reviewer for giving positive feedback on our research article.

Reviewer 3 Report

I attach some of my comments:

The contribution should not go in the affiliations. There is a specific section.
Certain things appear in red, I imagine it comes from some revision.
There are many formatting errors, please check it (Ex: extra space on line 7)
Line 44 alone.
I don't know if the format of the references is correct
The author contribution part is not justified
Figure 4 for example is not in bold.
Some titles end with a dot and some don't.
And a large number of formatting errors.
The number of decimal places used is each one.

It seems that in figure 1 there are bars (which are not appreciated) between the different treatments, others are not. What is that? Confidence interval? Mean and standard error? What do the letters mean differences within the graph or between them? Tables and figures should be self-explanatory. Figure 1, for example, has no differentiation between A, B, C ...

Figure 4 is illegible. What does the plus minus after) There are no letters? There are no significant differences?

What are the objectives of the work and how do they relate to the conclusions?

Like what diet supplementation? Not diet? There is no evaluation of the chemical composition of the diet? The amount of dry matter and moisture in the diet does not appear.

Nothing is explained in statistics. What was used as a fixed effect? There are parameters that are% or other types of variables where the method that should be used is not only an anova.

Author Response

We thank the reviewer for giving valuable comments about our review paper which is very helpful to improve the quality of the presentation of the manuscript. We have gone through the whole manuscript according to reviewer suggestions and modified the same. All changes in the manuscript have been made with red color fonts. 

QUES: The contribution should not go in the affiliations. There is a specific section. Certain things appear in red, I imagine it comes from some revision. There are many formatting errors, please check it (Ex: extra space on line 7) Line 44 alone. I don't know if the format of the references is correct. The author contribution part is not justified Figure 4 for example is not in bold. Some titles end with a dot and some don't. A large number of formatting errors. The number of decimal places used is each one.

ANS: Thanks for the reviewer comments. Yes, it is a revised version submitted to Animals. Therefore, revised matters have been made with the red colour font. We have gone through the whole manuscript and reformatted it according to the reviewer suggestion. We have prepared a reference format using Endnote software with MDPI style. For, author contribution, we have mentioned in the last according to journal guideline and deleted unusual dots.

QUES: It seems that in figure 1 there are bars (which are not appreciated) between the different treatments, others are not. What is that? Confidence interval? Mean and standard error? What do the letters mean differences within the graph or between them? Tables and figures should be self-explanatory. Figure 1, for example, has no differentiation between A, B, C … Figure 4 is illegible. What does the plus minus after) There are no letters? There are no significant differences?

ANS: Thanks for the valuable suggestion. For an easy understanding purpose, we have provided all data in table format with statistical values. We analyzed significant between times at confidence level 0.05 levels using SPSS16 software with following parameters one-way ANOVA, included post hoc, Duncan and descriptive analysis parameters, P values of less than 0.05 were considered statistically significant. In addition, statistical differences were differentiated by different alphabets; it indicates abcdeAlphabets in the column indicate significant differences between incubation periods (P≤0.05) 

QUES: What are the objectives of the work and how do they relate to the conclusions?

ANS: In this paper, we studied the in-vitro degradation of a group of polyphenolic compounds in Hanwoo steer rumen juice. The phenolic acids and flavonoids were evaluated the degradation potential in rumen juices at different time intervals. Also, we evaluated the rumen microbiome present in the rumen juice which is involved in the potent fermentation of secondary metabolites in the ruminants. Our main aim is to find out the status (may undergo degradation or not) of plant secondary metabolites (PSM) in rumen juice invitro. Further in vivo studies are needed to determine the PSM conversion rate, determination of final products from native PSM when undergoing degradation or reforming new complex and the effect on the microbiome

QUES: Like what diet supplementation? Not diet? There is no evaluation of the chemical composition of the diet? The amount of dry matter and moisture in the diet does not appear.

ANS: We would like to give kind information; we concentrated mainly on phenolic acids and flavonoid status in rumen juice at different time intervals. We used animals for the collection of rumen juice only and conducted experiments in-vitro. We have provided the composition of diet in table 1

QUES: Nothing is explained in statistics. What was used as a fixed effect? There are parameters that are% or other types of variables where the method that should be used is not only an anova.

 ANS: We analyzed significant between times at confidence level 0.05 levels using SPSS16 software with following parameters one-way ANOVA, included post hoc, Duncan and descriptive analysis parameters, P values of less than 0.05 were considered statistically significant. In addition, statistical differences were differentiated by different alphabets; it indicates abcdeAlphabets in the column indicate significant differences between incubation periods (P≤0.05). Line numbers 187-193.

Reviewer 4 Report

I think that authors and reviewers did a good job on this manuscript and I therefore would recommend acceptance in current form.

Author Response

I think that authors and reviewers did a good job on this manuscript and I therefore would recommend acceptance in current form.

Thank you for your positive comments

Round 2

Reviewer 3 Report

The authors have improved the manuscript.

This manuscript is a resubmission of an earlier submission. The following is a list of the peer review reports and author responses from that submission.

Round 1

Reviewer 1 Report

I can understand you evaluated the in vitro degradation of a group of PSM in the rumen fluid, collected from Hanwoo steer samples.

Please check format in References. Especially Journal name. J. Dairy Sci. (Italic)

My specific comments are as follows,

L16; it is --- It is
L30; firmicutes, bacterodetes, actinobacteria and syngergistetes --- Firmicutes, Bacterodetes, Actinobacteria and Syngergistetes (Italic)
L33; in-vitro --- Italic
L46; medium[4].. --- medium [4].
L48; forages[5] . --- forages [5].
L58; mans[7]. --- mans [7].
L60; secondary plant metabolites --- plant secondary metabolites (PSM)
L62; feces[8]. --- feces [8].
L66; rumen[9]. --- rumen [9].
L70; production[10]. --- production [10].
L73; rumen[11]. --- rumen [11].
L94; Grobig-DC --- Is this product name ? If so you should write company name and place.
L107; CaCl2.2H2O --- Check format.
L109; Na2S.9H2O --- Check format.
L117; analyzed[14] --- analyzed [14]
L135; ously[16]. --- ously [16].
L157; TCC.The gDNA --- TCC. The gDNA
Figure 1. Caffeic acid --- (µg) ? Add unit.
Incubation (hr --- Incubation (hr)
L194; n=3.ND --- n=3. ND
L202, L208, L209, L214; in vitro, in vivo --- Italic.
Figure 3. p. gallate --- propyl gallate ?
L217; Ester derivative --- Why do you use bold font only this word ?
L227, L228, L229; in Italic for Scientific name
L230; Intestinimonas --- only I is not Italic
L232; men(Fig.2B) --- men (Fig.2B)
fluid .In --- fluid. In
I.butyriciproducens --- I. butyriciproducens
L233; C.massilliensis --- C. massilliensis
P.ruminicola --- P. ruminicola
Figure 4. (A); What does each vertical bar represent ?
L242; digestibility[18]. --- digestibility [18].
Plant secondary metabolites (PSM) --- PSM
L252; fluid[6]. --- fluid [6].
L257; C-QA --- What is this ?
L266; chromatograms[20].. --- chromatograms [20].
L279; acids[22]. --- acids [22].
L285; production[24]. --- production [24].
L286; CT and HT --- What are these ?
L298; duction[25]. --- duction [25].
L309; nants[27]. --- nants [27].
L312; fermentation[28]. --- fermentation [28].
L314; ruminants[29]. --- ruminants [29].
L316; vitro[30]. --- vitro [30]. --- [30]. not Italic
L318-320; In addition, the in ... as substrates. --- Do you have idea what is the reason ?
L324; [18]. --- not Italic
L327; substrates[31]. --- substrates [31].

Author Response

Reviewer#1 Comment#1: I can understand you evaluated the in vitro degradation of a group of PSM in the rumen fluid, collected from Hanwoo steer samples. Author’s Response: Thank you for your suggestion and comments. Comment#2: Please check format in References. Especially Journal name. J. Dairy Sci. (Italic) Author’s Response: Thank you for your suggestion and comments. My specific comments are as follows, Comment#3: L16; it is --- It is Author’s Response: Thank you. Edited. Comment#4: L30; firmicutes, bacterodetes, actinobacteria and syngergistetes --- Firmicutes, Bacterodetes, Actinobacteria and Syngergistetes (Italic) Author’s Response: Thank you. Edited. Comment#5: L33; in-vitro --- Italic Author’s Response: Thank you. Edited. Comment#6: L46; medium[4].. --- medium [4]. Author’s Response: Thank you. Edited. Comment#7: L48; forages[5] . --- forages [5]. Author’s Response: Thank you. Edited. Comment#8: L58; mans[7]. --- mans [7]. Author’s Response: Thank you. Edited. Comment#9: L60; secondary plant metabolites --- plant secondary metabolites (PSM) Author’s Response: Thank you. Edited. Comment#10: L62; feces[8]. --- feces [8]. Author’s Response: Thank you. Edited. Comment#11: L66; rumen[9]. --- rumen [9]. Author’s Response: Thank you. Edited. Comment#12: L70; production[10]. --- production [10]. Author’s Response: Thank you. Edited. Comment#13: L73; rumen[11]. --- rumen [11]. Author’s Response: Thank you. Edited. Comment#14: L94; Grobig-DC --- Is this product name ? If so you should write company name and place. Author’s Response: Thank you. Edited. Comment#15: L110; CaCl2.2H2O --- Check format. Author’s Response: Thank you. Edited. Comment#16: L109; Na2S.9H2O --- Check format. Author’s Response: Thank you. Edited. Comment#17: L117; analyzed[14] --- analyzed [14] Author’s Response: Thank you. Edited. Comment#18: L135; ously[16]. --- ously [16]. Author’s Response: Thank you. Edited. Comment#19: L157; TCC.The gDNA --- TCC. The gDNA Author’s Response: Thank you. Edited. Comment#20: Figure 1. Caffeic acid --- (µg) ? Add unit. Incubation (hr --- Incubation (hr) Author’s Response: Thank you. Edited. Comment#21: L194; n=3.ND --- n=3. ND Author’s Response: Thank you. Edited. Comment#22: L202, L208, L209, L214; in vitro, in vivo --- Italic. Author’s Response: Thank you. Edited. Comment#22: Figure 3. p. gallate --- propyl gallate ? Author’s Response: Thank you. Edited. Comment#23: L217; Ester derivative --- Why do you use bold font only this word ? Author’s Response: Thank you. Edited. Comment#24: L227, L228, L229; in Italic for Scientific name Author’s Response: Thank you. Edited. Comment#25: L230; Intestinimonas --- only I is not Italic Author’s Response: Thank you. Edited. Comment#26: L232; men(Fig.2B) --- men (Fig.2B) fluid .In --- fluid. In I.butyriciproducens --- I. butyriciproducens Author’s Response: Thank you. Edited. Comment#27: L233; C.massilliensis --- C. massilliensis P.ruminicola --- P. ruminicola Author’s Response: Thank you. Edited. Comment#28: Figure 4. (A); What does each vertical bar represent ? Author’s Response: Thank you. Edited. Comment#29: L242; digestibility[18]. --- digestibility [18]. Plant secondary metabolites (PSM) --- PSM Author’s Response: Thank you. Edited. Comment#30: L252; fluid[6]. --- fluid [6]. Author’s Response: Thank you. Edited. Comment#31: L257; C-QA --- What is this ? Author’s Response: Thank you. Edited. Comment#32: L266; chromatograms[20].. --- chromatograms [20]. Author’s Response: Thank you. Edited. Comment#33: L279; acids[22]. --- acids [22]. Author’s Response: Thank you. Edited. Comment#34: L285; production[24]. --- production [24]. Author’s Response: Thank you. Edited. Comment#35: L286; CT and HT --- What are these ? Author’s Response: Thank you. Edited. Comment#36: L298; duction[25]. --- duction [25]. Author’s Response: Thank you. Edited. Comment#37: L309; nants[27]. --- nants [27]. Author’s Response: Thank you. Edited. Comment#38: L312; fermentation[28]. --- fermentation [28]. Author’s Response: Thank you. Edited. Comment#39: L314; ruminants[29]. --- ruminants [29]. Author’s Response: Thank you. Edited. Comment#40: L316; vitro[30]. --- vitro [30]. --- [30]. not Italic Author’s Response: Thank you. Edited. Comment#41: L318-320; In addition, the in ... as substrates. --- Do you have idea what is the reason ? Author’s Response: Thank you. Edited. Comment#42: L324; [18]. --- not Italic Author’s Response: Thank you. Edited. Comment#43: L327; substrates[31]. --- substrates [31]. Author’s Response: Thank you. Edited.

Reviewer 2 Report

Review of the manuscript “Microbial dynamics and in-vitro degradation of plant secondary metabolites in Hanwoo steer rumen fluids

General comments

The subject of the paper is in agreement with the scope of the Journal. 

The objective of the experiment is very interesting.

However, the paper presents some deficiencies:

From the methodological point of view

1.1 The description of the incubation is not clear. Were the plant secondary metabolites (PSM) incubated all together, as reported in the abstract (line 22), or separately prepared and added to the fermentation vessel, as reported at lines 124-126. How many samples of each PSM were incubated? Why the temperature was 37°C and not 39°C, as suggested by the company (Ankom Technology Corporation)?

1.2 The statistical analysis is not well described. One-way ANOVA is reported but it is not explained the main factor (phenolic acids and flavonoids or incubation times?).

1.3 The pattern of the degradation during the incubation is not well described from the statistical point of view (i.e. the significance of the linearly, quadratic effect etc.)

From the scientific point of view

2.1 The paper is very interesting but, in my opinion, some important parameters of the fermentation (i.e. pH, volatile fatty acids, N-NH3) are missing. The measurements of these parameters at the end of the incubation could have provided information on the effects of plant secondary metabolites in the rumen environment.

2.2 The interaction among different PSM is not considered in this paper but I think that a brief discussion on these aspects could be interesting. 

Specific comments

Line 16              in the rumen. It is possible…

Line 23              see M&M. The normal temperature used in the Daisy system is 39°C (not 37°C)

Line 23              see M&M. Were the PSM all incubated together? The mixture of selected PSM was incubated at 37 °C for 12 – 72 h,

Lines 23-24        see M&M. On what basis were the incubation times chosen?

Line 39 F            orages contain not only…

Line 46              …fermentation medium[4]. In contrast

Line 88              …around 580 ± s.d. kg (30 ± s.d. months old)

Lines 89-90        “Hanwoo steer received a diet of Italian ryegrass (Lolium multiflorum) at 2 kg per day ad libitum” It is not clear if the quantity of Italian ryegrass was fixed (2 kg/d) or ad libitum

Lines 90-91        The concentrate and Italian ryegrass were 90 present at a ratio of 3:7. If the quantity of Italian ryegrass is fixed at 2 kg/d, the quantity of concentrate is 0.857 kg/d. But on Table 1 is described the diet supplementation and the concentrate represents the percentage of 75% of the total. And rice and timothy?? It is important to better explain the composition and formulation of the diet.

Lines 91-94        If the formulation of the concentrate is shown in Table 1, it is not necessary to report the list of ingredients and the percentages in the text.

Table 1              See the guidelines for Authors (no bold for ingredients etc.)

Table 1              I suggest indicating as the last ingredient “vitamin and mineral mix” and not the commercial name of the supplement.

Line 106            I suggest indicating a more recent reference instead of Goering and Van Soest (1970). For example, Holden (1999), J. Dairy Sci.

Line 121 and Line 129 

                          I have serious doubts about the temperature (37°C) adopted for the incubation in Ankom Technology Daisy II® incubator. The Instruction suggested by the company (Ankom Technology Corporation) indicates an optimal value of 39°C. This methodological change is not explained 

Line 122-123     …. the incubator allows for the simultaneous incubation of up 122 to 11 different phenolic and flavonoid samples placed in glass containers…. Why 11?? The selected PSM are 10 (rutin, vitexin, myricetin, p-coumaric acid, ferulic acid, caffeic acid, quercetin, luteolin, propyl gallate, and kaempferol). It is not clear the adopted procedure (number of samples, number of replication, number of runs….)

Line 122-123     It is not clear if the PSM were incubated together or separately. In the abstract, a mixture of selected PMS is reported but in M&M we find “Each pure 124 phenolic and flavonoid (1 mg) sample was separately prepared and added to the fermentation vessel containing buffered inoculums.

Lines 164-167    The ruminal fermentation and microbial population data were analyzed with one way ANOVA” What is the statistical model adopted? What is the experimental factor of the one-way analysis?

  1. PMS effect: we have 10 different PMS (levels) but probably all PMS are incubated together? How many replications have been considered?
  2. Incubation time effect: we have 5 different incubation times (levels), so 4 degrees of freedom. The significance of the linear, quadratic cubic… components can be calculated.

Lines 166-167    The differences between the means (??) were analyzed using Duncan's multiple range test. No significance of differences (P-value) between means has been reported in the section of Results. So I conclude that the results are similar.  

Line 173             During the first 12h incubation, coumaric acid, and ferulic were started to disappear its concentrations significantly (P=????)

Line 181, 194, 220, 239 The data were represented as Mean± STD, n=3.

Figures 1, 2, 3    The data were represented as Mean± STD, n=3. No standard errors have been reported in these Figures.

Figures 1, 2, 3    To verify the differences among the incubation times the use of ***, **, * or different letters can be used for each bar in the graphs.

Lines 203-215    This part can be moved to the discussion (?)

Line 230, 232, 234 Figure

Line 234            Figure 2C

Author Response

Response to Reviewer#2

Comments#1 General comments. The subject of the paper is in agreement with the scope of the Journal. The objective of the experiment is very interesting. However, the paper presents some deficiencies:

Author’s Responses: Thank you for your comments and suggestions.

Comments#2 From the methodological point of view

1.1 The description of the incubation is not clear. Were the plant secondary metabolites (PSM) incubated all together, as reported in the abstract (line 22), or separately prepared and added to the fermentation vessel, as reported at lines 124-126. How many samples of each PSM were incubated? Why the temperature was 37°C and not 39°C, as suggested by the company (Ankom Technology Corporation)?

Author’s Responses: The experimental parts clearly mentioned in the revised version.

Comments#3 1.2 The statistical analysis is not well described. One-way ANOVA is reported but it is not explained the main factor (phenolic acids and flavonoids or incubation times?).

Author’s Responses: Statistical data checked and added in the revised manuscript. Thank you

Comments#4 1.3 The pattern of the degradation during the incubation is not well described from the statistical point of view (i.e. the significance of the linearly, quadratic effect etc.)

Author’s Responses: In this communication, we evaluated preliminary study of the metabolites degradation using HPLC-DAD methods. In addition, we were planned to evaluate the in-vivo study that could be included clearly the detail mechanism and metabolites biotransformation in in-vitro and in-vivo. Thank you

Comments#5 From the scientific point of view

2.1 The paper is very interesting but, in my opinion, some important parameters of the fermentation (i.e. pH, volatile fatty acids, N-NH3) are missing. The measurements of these parameters at the end of the incubation could have provided information on the effects of plant secondary metabolites in the rumen environment.

Author’s Responses: Edited.

Comments#6 2.2 The interaction among different PSM is not considered in this paper but I think that a brief discussion on these aspects could be interesting. 

Author’s Responses: Added content from available recent literatures. Thank you

Specific comments

Comments#7 Line 16 in the rumen. It is possible…

Author’s Responses: Edited.

Comments#8 Line 23 see M&M. The normal temperature used in the Daisy system is 39°C (not 37°C)

Author’s Responses: Edited.

Comments#9 Line 23 see M&M. Were the PSM all incubated together? The mixture of selected PSM was incubated at 37 °C for 12 – 72 h,

Author’s Responses: The information corrected in the revised manuscript. Thank you

Comments#10 Lines 23-24 see M&M. On what basis were the incubation times chosen?

Author’s Responses: As per the previous literature survey we carried out the experimental set up with slight modifications.

Comments#11 Line 39 Forages contain not only…

Author’s Responses: Thank you. Edited

Comments#12 Line 46 …fermentation medium[4]. In contrast

Author’s Responses: Thank you. Edited

Comments#13 Line 88   …around 580 ± s.d. kg (30 ± s.d. months old)

Author’s Responses: The statistical information included in the revised manuscript.

Comments#14 Lines 89-90“Hanwoo steer received a diet of Italian ryegrass (Lolium multiflorum) at 2 kg per day ad libitum” It is not clear if the quantity of Italian ryegrass was fixed (2 kg/d) or ad libitum

Author’s Responses: The methodology section revised as per your comments. Now it can be readable.

Comments#15 Lines 90-91 The concentrate and Italian ryegrass were 90 present at a ratio of 3:7. If the quantity of Italian ryegrass is fixed at 2 kg/d, the quantity of concentrate is 0.857 kg/d. But on Table 1 is described the diet supplementation and the concentrate represents the percentage of 75% of the total. And rice and timothy?? It is important to better explain the composition and formulation of the diet.

Author’s Responses: Edited. The formulation diet clearly mentioned in the revised manuscript. Thank you

Comments#16 Lines 91-94 If the formulation of the concentrate is shown in Table 1, it is not necessary to report the list of ingredients and the percentages in the text.

Author’s Responses: Yes. We agreed and edited in the revised manuscript. Thank you.

Comments#17 Table 1 See the guidelines for Authors (no bold for ingredients etc.)

Author’s Responses: Sorry. We now carefully edited as per the journal author guidelines.

Comments#18 Table 1 I suggest indicating as the last ingredient “vitamin and mineral mix” and not the commercial name of the supplement.

Author’s Responses: We agreed your suggestion and table 1 edited in the revised manuscript.

Comments#19 Line 106 I suggest indicating a more recent reference instead of Goering

and Van Soest (1970). For example, Holden (1999), J. Dairy Sci.

Author’s Responses: We agreed your suggestion and table 1 edited in the revised manuscript.

Comments#20 Line 121 and Line 129 

I have serious doubts about the temperature (37°C) adopted for the incubation in Ankom Technology Daisy II® incubator. The Instruction suggested by the company (Ankom Technology Corporation) indicates an optimal value of 39°C. This methodological change is not explained 

Author’s Responses: Thank you. Edited in the revised manuscript. Now it can be read clearly in the revised manuscript.

Comments#21 Line 122-123     …. the incubator allows for the simultaneous incubation of up 122 to 11 different phenolic and flavonoid samples placed in glass containers…. Why 11?? The selected PSM are 10 (rutin, vitexin, myricetin, p-coumaric acid, ferulic acid, caffeic acid, quercetin, luteolin, propyl gallate, and kaempferol). It is not clear the adopted procedure (number of samples, number of replication, number of runs….)

Author’s Responses: Thank you. Edited in the revised manuscript. Now it can be read clearly in the revised manuscript.

Comments#22 Line 122-123     It is not clear if the PSM were incubated together or separately. In the abstract, a mixture of selected PMS is reported but in M&M we find “Each pure 124 phenolic and flavonoid (1 mg) sample was separately prepared and added to the fermentation vessel containing buffered inoculums.

Author’s Responses: Thank you. Edited in the revised manuscript. Now it can be read clearly in the revised manuscript.

Comments#23 Lines 164-167    The ruminal fermentation and microbial population data were analyzed with one way ANOVA” What is the statistical model adopted? What is the experimental factor of the one-way analysis?

  1. PMS effect: we have 10 different PMS (levels) but probably all PMS are incubated together? How many replications have been considered?
  2. Incubation time effect: we have 5 different incubation times (levels), so 4 degrees of freedom. The significance of the linear, quadratic cubic… components can be calculated.

Author’s Responses: Thank you. Edited in the revised manuscript. Now it can be read clearly in the revised manuscript.

Comments#24 Lines 166-167    The differences between the means (??) were analyzed using Duncan's multiple range test. No significance of differences (P-value) between means has been reported in the section of Results. So I conclude that the results are similar.  

Author’s Responses: The statistical information added in the revised manuscript. Now, the revised manuscript has reported statistical significance value between control and treated groups. 

Comments#25 Line 173 During the first 12h incubation, coumaric acid, and ferulic were started to disappear its concentrations significantly (P=????)

Author’s Responses: The statistical evolutions of the obtained numerical values were clearly mentioned in the revised manuscript. Thank you

Comments#26 Line 181, 194, 220, 239 The data were represented as Mean± STD, n=3.

Figures 1, 2, 3    The data were represented as Mean± STD, n=3. No standard errors have been reported in these Figures.

Author’s Responses: The statistical were stated clearly in the revised manuscript. Thank you

Comments#27 Figures 1, 2, 3    To verify the differences among the incubation times the use of ***, **, * or different letters can be used for each bar in the graphs.

Author’s Responses: Edited. Thank you

Comments#28 Lines 203-215    This part can be moved to the discussion (?)

Author’s Responses: Edited. The mentioned content moved to discussion section. Thank you

Comments#29 Line 230, 232, 234 Figure

Author’s Responses: Edited. Thank you

Comments#30 Line 234 -Figure 2C

Author’s Responses: Edited. Thank you

Reviewer 3 Report

Dear Authors,

The manuscript entitled "Microbial dynamics and in-vitro degradation of plant secondary metabolites in Hanwoo steer rumen fluids" has been reviewed.
I have serious concerns that drive me to do not support the publication of this study in Animals Journal.

Specifically:

Overall, the editing is bad and requires extensive revision (double dots, spaces, etc). It gives the sense of incomplete work.

Animals journal required a simple summary, this was not provided at all.

Institutional Review Board Statement is obligatory since you used fistulated steers. 

Line 16: I can't understand if you want two or one sentence. Please rewrite.

Keywords should be different from the title in order to expand article views. In addition, they should be presented in alphabetical order. 

Line 29-31: I can't consider this statement as a result of your study since these phyla are the dominants in every rumen fluid. 

line 36: Why? How? Reference? What is beneficial for ruminant nutrition? 

Line 37-39: Are the high protein degradation and ammonia release valuable within the rumen? I doubt. 

Line 36: Double dot.

Although the introduction has high scientific soundness and references, it is very wordy and creates circles. The introduction should be like a funnel, you should start from a general observation (ex. plant secondary metabolites), then you should state the negatives and positives of their supplementation in ruminants, then you should provide information about rumen microbiome and PSM, and what has been investigated up to now in this topic. After, you should clearly provide which is the novelty and state of art of your research and state your objective. The introduction already has most of the above information, however, are not consistent.

Line 89: at 2 kg per day ad libitum…2 kg or ad libitum?
The diet is not clearly presented. Please provide exactly the quantities instead of the ratio between forage and concentrates.

There is no information about the samples of 3 donors. Did you combine the rumen fluids from different donors to one daisy vessel?

The 13th citation refers to an experiment (from the same corresponding author) that run between 2008-2010. If you collected the rumen fluid in 2010, then when did you run the in vitro analyses or even more the metagenomic analysis. Where did you store the rumen fluid and for how many years?

Why did you choose to ferment the rumen at 37 instead of 39? Please provide justification.

Line 131:At each time, the rumen fermented samples were carefully collected using a 5 ml sterile syringe to evaluate the hydrolysis of the polyphenolic compounds. Did you make any homogenization before you collect the 5 ml sample? Since the 1 mg is a very small amount, a proper representative sample collection was obligatory. 

Line 153: Not machine but a device.

In my opinion, further information should be provided for metagenomic analysis such OTU etc. In addition, it is not clear if you used a 454 pyrosequencing or illumina. There are controversial sentences (the same in 3.4. chapter). 

In material and methods, there is no clear information for number of samples fermented and analyzed in each daisy vessel. 

Line 183: Do you mean represented?

I can’t observe the STD in figure 1. Maybe it is small (STD) but absolutely the colors are not appropriate. 
N=3 is not a representative sample.
In figure 1c there is no unit for caffeic acid.

In figure 1 there is no statistical information for differences. You should provide the P-value for each graphical and superscripts letter to depict alterations between time.

Figures 2 and 3 portrayed the same biases as those of figure 1.

I struggle to understand what was the objective of rumen microbiome analysis. There is no explanation for the 3 samples. Did you collect rumen fluid samples for DNA extraction before or after (and in which interval) in vitro fermentation? 

In figure 4a you stated n=6, while in the figure's legend there is again the n=3. I am confused. 

Considering the STD of prevotella (Figure 4b), I can suppose that there was a rumen fluid sample with almost 0 % prevotella, is this possible by taking into account that prevotella appears to be the predominant genus in the rumen? 

The discussion does not provide assumptions for research findings and there is no justified linkage between PSM degradation and rumen microbiota.

Overall, this study investigates a "hot" trend in ruminant nutrition using high edge techniques. On my first look, I was excited to read this manuscript. However, I struggle to understand the experimental design and the entire conceptualization since there was not any connection between PSM degradation and the rumen microbiome. The study in the current form is misleading and the readers are unable to extract certain conclusions. I am very sad that I can not support the publication of this manuscript in Animals journal. However, I encourage you to consider the followings and resubmit a revised original article instead of communication:

1) Change the whole introduction using the same references you had but following a straightforward pipe-line leading to your objective.

2) Provide a material and methods section with a detailed explanation of experimental design (number of samples, diet composition and quantities in donor cows, sequencing assay, storage conditions etc). You should avoid any comment on material and methods in the next submission.

3) Results should be presented providing the means and significance. Be precise and consistent. Group the genus and species by comparable abundances in each figure in order to be clear for the readers. 

4) Discuss your results using the latest published studies on the topic without being wordy.

Author Response

Reviewer#3

Comment#1 The manuscript entitled "Microbial dynamics and in-vitro degradation of plant secondary metabolites in Hanwoo steer rumen fluids" has been reviewed. I have serious concerns that drive me to do not support the publication of this study in Animals Journal.
Author’s Response: Thank you for your comment. We modified and improves the whole manuscript as you suggested

Specifically:

Comment#2 Overall, the editing is bad and requires extensive revision (double dots, spaces, etc). It gives the sense of incomplete work.

Author’s Response: Thank you. The whole text is edited as you recommended

Comment#3 Animals journal required a simple summary, this was not provided at all.

Institutional Review Board Statement is obligatory since you used fistulated steers. 

Author’s Response: Thank you. We agreed and animal board committee statement added in the revised manuscript

Comment#4 Line 16: I can't understand if you want two or one sentence. Please rewrite.

Author’s Response: Thank you. The sentence edited ad you suggested.

Comment#5 Keywords should be different from the title in order to expand article views. In addition, they should be presented in alphabetical order. 

Author’s Response: Thank you for your comment. Keywords edited in the revised manuscript.

Comment#6 Line 29-31: I can't consider this statement as a result of your study since these phyla are the dominants in every rumen fluid. 

Author’s Response: Thank you. Edited in the revised manuscript. Now it can be read clearly in the revised manuscript.

Comment#7 line 36: Why? How? Reference? What is beneficial for ruminant nutrition? 

Author’s Response: Thank you. Edited in the revised manuscript. Now it can be read clearly in the revised manuscript.

Comment#8 Line 37-39: Are the high protein degradation and ammonia release valuable within the rumen? I doubt. 

Author’s Response: Thank you. Edited in the revised manuscript. Now it can be read clearly in the revised manuscript.

Comment#9 Line 36: Double dot.

Author’s Response: Thank you. Edited in the revised manuscript. Now it can be read clearly in the revised manuscript.

Comment#10 Although the introduction has high scientific soundness and references, it is very wordy and creates circles. The introduction should be like a funnel, you should start from a general observation (ex. plant secondary metabolites), then you should state the negatives and positives of their supplementation in ruminants, then you should provide information about rumen microbiome and PSM, and what has been investigated up to now in this topic. After, you should clearly provide which is the novelty and state of art of your research and state your objective. The introduction already has most of the above information, however, are not consistent.
Author’s Response: Thank you. Edited in the revised manuscript. Now it can be read clearly in the revised manuscript.

Comment#11 Line 89: at 2 kg per day ad libitum…2 kg or ad libitum?
The diet is not clearly presented. Please provide exactly the quantities instead of the ratio between forage and concentrates.

Author’s Response: Thank you. Edited in the revised manuscript. Now it can be read clearly in the revised manuscript.

Comment#12 There is no information about the samples of 3 donors. Did you combine the rumen fluids from different donors to one daisy vessel?

Author’s Response: Thank you. Edited in the revised manuscript. Now it can be read clearly in the revised manuscript.

Comment#13 The 13th citation refers to an experiment (from the same corresponding author) that run between 2008-2010. If you collected the rumen fluid in 2010, then when did you run the in vitro analyses or even more the metagenomic analysis. Where did you store the rumen fluid and for how many years?

Author’s Response: Thank you. Edited in the revised manuscript. Now it can be read clearly in the revised manuscript.

Comment#14 Why did you choose to ferment the rumen at 37 instead of 39? Please provide justification.

Author’s Response: Thank you. Edited in the revised manuscript. Now it can be read clearly in the revised manuscript.

Comment#15 Line 131:At each time, the rumen fermented samples were carefully collected using a 5 ml sterile syringe to evaluate the hydrolysis of the polyphenolic compounds. Did you make any homogenization before you collect the 5 ml sample? Since the 1 mg is a very small amount, a proper representative sample collection was obligatory.

Author’s Response: Thank you. Edited in the revised manuscript. Now it can be read clearly in the revised manuscript.

Comment#16 Line 153: Not machine but a device.

In my opinion, further information should be provided for metagenomic analysis such OTU etc. In addition, it is not clear if you used a 454 pyrosequencing or illumina. There are controversial sentences (the same in 3.4. chapter).

Author’s Response: Thank you. Edited in the revised manuscript. Now it can be read clearly in the revised manuscript.

Comment#17 In material and methods, there is no clear information for number of samples fermented and analyzed in each daisy vessel.

Author’s Response: Thank you. Edited in the revised manuscript. Now it can be read clearly in the revised manuscript.

Comment#18 Line 183: Do you mean represented?

I can’t observe the STD in figure 1. Maybe it is small (STD) but absolutely the colors are not appropriate. 
N=3 is not a representative sample.

Author’s Response: Thank you. Edited in the revised manuscript. Now it can be read clearly in the revised manuscript.

Comment#19 In figure 1c there is no unit for caffeic acid.

Author’s Response: Thank you. Edited in the revised manuscript. Now it can be read clearly in the revised manuscript.

Comment#20 In figure 1 there is no statistical information for differences. You should provide the P-value for each graphical and superscripts letter to depict alterations between time.

Author’s Response: Thank you. Edited in the revised manuscript. Now it can be read clearly in the revised manuscript.

Comment#21 Figures 2 and 3 portrayed the same biases as those of figure 1.

I struggle to understand what was the objective of rumen microbiome analysis. There is no explanation for the 3 samples. Did you collect rumen fluid samples for DNA extraction before or after (and in which interval) in vitro fermentation? 

Author’s Response: Thank you. Edited in the revised manuscript. Now it can be read clearly in the revised manuscript.

Comment#22 In figure 4a you stated n=6, while in the figure's legend there is again the n=3. I am confused. 

Author’s Response: Thank you. Edited in the revised manuscript. Now it can be read clearly in the revised manuscript.

Comment#23 Considering the STD of prevotella (Figure 4b), I can suppose that there was a rumen fluid sample with almost 0 % prevotella, is this possible by taking into account that prevotella appears to be the predominant genus in the rumen? 

Author’s Response: Thank you. Edited in the revised manuscript. Now it can be read clearly in the revised manuscript.

Comment#24 The discussion does not provide assumptions for research findings and there is no justified linkage between PSM degradation and rumen microbiota.

Author’s Response: Thank you for your comment. The discussion clearly stated with previous literatures reports. Not added any vague or un-authenticate statements.

Comment#25 Overall, this study investigates a "hot" trend in ruminant nutrition using high edge techniques. On my first look, I was excited to read this manuscript. However, I struggle to understand the experimental design and the entire conceptualization since there was not any connection between PSM degradation and the rumen microbiome. The study in the current form is misleading and the readers are unable to extract certain conclusions. I am very sad that I can not support the publication of this manuscript in Animals journal. However, I encourage you to consider the followings and resubmit a revised original article instead of communication:

Author’s Response: The whole manuscript revised as you commended. Now it can be more clear and good presentation. Thank you

Comment#26 1) Change the whole introduction using the same references you had but following a straightforward pipe-line leading to your objective.

Author’s Response: We agreed and modified the introduction section in the revised manuscript. Thank you

Comment#27 2) Provide a material and methods section with a detailed explanation of experimental design (number of samples, diet composition and quantities in donor cows, sequencing assay, storage conditions etc). You should avoid any comment on material and methods in the next submission.

Author’s Response: the materials and methods section revised as you pointed out. Thank you

Comment#28 3) Results should be presented providing the means and significance. Be precise and consistent. Group the genus and species by comparable abundances in each figure in order to be clear for the readers. 

Author’s Response: We agreed your comments and edited accordingly. Now it can be read in the revised manuscript. Thank you

Comment#29 4) Discuss your results using the latest published studies on the topic without being wordy.
Author’s Response: Thank you for your suggestion. We updated discussion part with recent literatures.

Round 2

Reviewer 2 Report

The required changes have been accepted by the Authors in the answers report but the manuscript has been not modified (i.e. the paragraph of the statistical analysis is the same of that of the original manuscript, no statistical model is reported) (some important parameters of the fermentation (i.e. pH, volatile fatty acids, N-NH3) are missing)

No answers have been given regarding some requests for explanations on the methodological approach (e.g. why did the authors work at 37°C and not at 39°C as indicated in the guidelines of the Ankom instruments?)

The corrections reported in the revised manuscript are only formal but not substantive, as required. 

Author Response

Response to Reviewer#2

Comment#1

The required changes have been accepted by the Authors in the answers report but the manuscript has been not modified

(i.e. the paragraph of the statistical analysis is the same of that of the original manuscript, no statistical model is reported)

(some important parameters of the fermentation (i.e. pH, volatile fatty acids, N-NH3) are missing)

Author’s Response: Thank you very much for your concern again in our manuscript. Certainly, we are conducted pH parameters after addition of phenolic acids and flavonoids in the buffer medium. The initial pH was 6.7 and end of experimental period was 6.4 (therefore, no significant changes were observed in control and experimental samples- data not included in this study). Also, we concentrated mainly on phenolic acids and flavonoids profile in this paper. We understood that volatile fatty acids, N-NH3 is essential parameters to analysis that you recommended too. In addition, we are minded and do it in our future work.

Comment#2

No answers have been given regarding some requests for explanations on the methodological approach (e.g. why did the authors work at 37°C and not at 39°C as indicated in the guidelines of the Ankom instruments?)

Author’s Response: Yes, thank you very much for your query again and sorry for the missed to address the reviewer query. 1) In this experiment we optimized the temperature parameters and as lower is better performance for the degradation of some plant based compounds.

2) Also, we stated in the methodology section that slightly modified methods in this study because we conducted longer duration analysis (0-72 hr) of metabolites degradation. Therefore, the temperatures parameters slightly modified in this experiments.

Comment#3

The corrections reported in the revised manuscript are only formal but not substantive, as required. 

Author’s Response: The revised manuscript is substantially improved and modified/ corrected all reviewer suggestions and comments in the revised manuscript.

Reviewer 3 Report

Dear authors,

Unfortunately, almost 90% of my recommendations were not addressed. It is understandable that within a revision process not all reviewers’ concerns may be fulfilled. However, in your case, you did not even provide a short justification based on your point of view. Instead, you responded 18 times as
"Thank you. Edited in the revised manuscript. Now it can be read clearly in the revised manuscript." Animals’ guidelines clearly describe the revision process; amongst the guidelines, you may observe: "The author needs to provide a point by point response or provide a rebuttal if some of the reviewer’s comments cannot be revised. Usually, only one round of major revisions is allowed. Authors will be asked to resubmit the revised paper within a suitable time frame, and the revised version will be returned to the reviewer for further comments. (Please detail the revisions that have been made, citing line number and exact change, so that the editor can check the changes expeditiously. Simple statements like ‘done’ or ‘revised as requested’ will not be accepted, unless the change is simply a typographical error)". 
Putting aside the scientific perspective, your refusal to counteract the revision process degrades the entire peer-review procedure and reviewers' time and effort. 

Nevertheless, I will once again summarize my major concerns:

1) I had suggested a thorough revision of the introduction due to a lack of consistency. However. the introduction section in the revised manuscript depicts changes in only 8 words. The 7 of them are missing spaces.

2) Line 89-92: In addition, “The concentrate contained 30.49% maize, 15.56% corn gluten feed, 7.37% soybean meal, 10.01% wheat gluten, 3.12% rapeseed meal, 3.11% coconut kernel meal, 3.11% palm kernel meal, 1.33% limestone, 0.36% salt, 0.36% sodium bicarbonate, and 0.21% Grobig-DC”.

The sum is 75.03 %! Why should readers have to scroll down to realize that you have also added rice straw?

3) Reviewer: The 13th citation refers to an experiment (from the same corresponding author) that ran between 2008-2010. If you collected the rumen fluid in 2010, then when did you run the in vitro analyses or more so the metagenomic analysis. Where did you store the rumen fluid and for how many years?

No explanation was given. 

4) Reviewer: Line 89: at 2 kg per day ad libitum…2 kg or ad libitum?
The diet is not clearly presented. Please provide exactly the quantities instead of the ratio between forage and concentrates.

Author’s Response: Thank you. Edited in the revised manuscript. Now it can be read clearly in the revised manuscript.

“During the experimental period, each Hanwoo steer received a diet of Italian ryegrass (Lolium multiflorum) at 2 kg per day with free access to water (NIAS 2007). The concentrate and Italian ryegrass were present at a ratio of 3:7.”

How is it possible to feed a 580 kg steer with 2 kg per day Lolium multiflorum and 0.85 kg concentrate (based on your ratio 7:3) ?

5) Refusal to justify the incubation setup at 37 oC instead of 39oC despite the fact that this was pointed out by 2 reviewers.

Author Response

Response to Reviewer#3

 Comment#1

Unfortunately, almost 90% of my recommendations were not addressed. It is understandable that within a revision process not all reviewers’ concerns may be fulfilled. However, in your case, you did not even provide a short justification based on your point of view. Instead, you responded 18 times as "Thank you. Edited in the revised manuscript. Now it can be read clearly in the revised manuscript.Animals’ guidelines clearly describe the revision process; amongst the guidelines, you may observe: "The author needs to provide a point by point response or provide a rebuttal if some of the reviewer’s comments cannot be revised. Usually, only one round of major revisions is allowed. Authors will be asked to resubmit the revised paper within a suitable time frame, and the revised version will be returned to the reviewer for further comments. (Please detail the revisions that have been made, citing line number and exact change, so that the editor can check the changes expeditiously. Simple statements like ‘done’ or ‘revised as requested’ will not be accepted, unless the change is simply a typographical error)". 
Putting aside the scientific perspective, your refusal to counteract the revision process degrades the entire peer-review procedure and reviewers' time and effort. 
Nevertheless, I will once again summarize my major concerns:

1) I had suggested a thorough revision of the introduction due to a lack of consistency. However
, the introduction section in the revised manuscript depicts changes in only 8 words. The 7 of them are missing spaces. 

Author’s Response: Sorry for the inconvenience caused. Now, we carefully read all reviewer comments/suggestions and corrected accordingly.

Line Numbers: 38-45. The interest of the research on the effects polyphenols on animal nutrition is significant increased. These compounds are ubiquitous in the plant kingdom particularly in plants exploited as feeding resources for alternative to cultivated crops and many agro- industrial by-products. Among plant secondary metabolites, various polyphenols are found, ranging from simple phenolic molecules to highly polymerized compounds. This has been modulating ruminal metabolism[2], growth performance, health status[3], and product quality[4].

Line Numbers: 58-67 Sinz et al. (2018) reported the group of flavonoids such as tannic acid, epicatechin, quercetin, isoquercetin, and luteolin-7-glucoside reduced the amount of methane produced in relation to total gas produced.  From the flavonoids tested especially luteolin-7-glucoside seems to have a similar potential as tannic acid to mitigate methane and ammonia formation during ruminal fermentation in vitro, both favourable in environmental respect. Also, Kong et al., (2019) recently studied the probiotics or the combination of yeast and phytochemicals mainly flavonoids could be compatible with each other improved rumen fermentation, but supplementation with C. tropicalis had limited effects on increasing growth performance and decreasing fecal scores compared with flavonoid supplementation. Therefore, the flavonoids effect on health or rumen fermentation in pre- and post-weaning calves (Kong et al., 2019) (Line No: 47-56)

Comment#2

2) Line 89-92: In addition, “The concentrate contained 30.46% maize, 15.56% corn gluten feed, 7.37% soybean meal, 10.01% wheat gluten, 3.12% rapeseed meal, 3.11% coconut kernel meal, 3.11% palm kernel meal, 1.33% limestone, 0.36% salt, 0.36% sodium bicarbonate, and 0.21% Grobig-DC” and remaining 25% were forages

The sum is 75.03 %! Why should readers have to scroll down to realize that you have also added rice straw?

Author’s Response: Yes. Thank you for reviewer query. Total concentrate is 75%) and Forages is 25%, which includes rice straw and timothy feed given for the experimental animals). This feed contains potent selected phenolics group, hence, we fed animals with these silage feed samples for throughout experimental period.

Comment#3

3) Reviewer: The 13th citation refers to an experiment (from the same corresponding author) that ran between 2008-2010. If you collected the rumen fluid in 2010, then when did you run the in vitro analyses or more so the metagenomic analysis. Where did you store the rumen fluid and for how many years?

No explanation was given.

Author’s Response: The methodology and feed composition ratio were adapted from our previous study. Furthermore, the rumen fluid collection and study was conducted in November 2019- Octobere 2020. In addition, the collected rumen fluid samples were stored at -80◦C for 3 weeks for metagenomic analysis and pyro-sequence carried out at Macrogen Company Seoul, Korea.

Comment#4

4) Reviewer: Line 89: at 2 kg per day ad libitum…2 kg or ad libitum?
The diet is not clearly presented. Please provide exactly the quantities instead of the ratio between forage and concentrates.

Author’s Response:  Yes agreed with reviewer comment. Actually, 12kg of diet were supplemented to each animal which consist of 2.5% forages and 7.5% concentrate and also has a mistake, we did not use

“During the experimental period, each Hanwoo steer received a diet of Italian ryegrass (Lolium multiflorum) at 2 kg per day with free access to water (NIAS 2007). The concentrate and Italian ryegrass were present at a ratio of 3:7.”

How is it possible to feed a 580 kg steer with 2 kg per day Lolium multiflorum and 0.85 kg concentrate (based on your ratio 7:3) ?

Author’s Response: It was great mistake did by our researcher. Actually, we had provided 12kg of total diets / steer/ day, it consist of both forages and concentrate in the ratio of 2.5:7.5, respectively. We received rumen juice from other farm unit, and we collected information’s about feeding system, it gives some misunderstanding between farm person and our researcher. We mainly targeted our work to get the information about whether polyphenolic compounds have been degradation/ changed or not in rumen under in-vitro conditions, because this experiment mimics like in- vivo model.

Comment#5

5) Refusal to justify the incubation setup at 37 oC instead of 39oC despite the fact that this was pointed out by 2 reviewers.

Author’s Response: Yes, thank you very much for your query again and sorry for the missed to address the reviewer query. 1) In this experiment we optimized the temperature parameters and as lower is better performance for the degradation of some plant based compounds.

2) Also, we stated in the methodology section that slightly modified methods in this study because we conducted longer duration analysis (0-72 hr) of metabolites degradation. Therefore, the temperatures parameters slightly modified in this experiments.